# DrugAgent: Multi-Agent Large Language Model-Based Reasoning for Drug-Target Interaction Prediction and Repurposing

## Abstract

Advancements in large language models (LLMs) allow them to address a wide set of questions from diverse topics using human-like text interfaces, but limitations in their training prevent them from answering accurately in scenarios that could benefit from multiple perspectives. Multi-agent systems allow the resolution of questions to enhance result consistency and reliability. Here we create a multi-perspective (i.e., unstructured text, structured knowledge graph, and Machine Learning (ML) prediction) multi-agent LLM system. We apply this system to the biologically inspired problem of predicting drug-target interaction. Our system uses a coordinator agent to assign and integrate results for tasks given to three specialized agents: an AI agent for ML predictions, a knowledge graph (KG) agent for KG retrieval, and a search agent for web-based information retrieval.

We conducted experiments using our LLM-based system for predicting drug-target interaction constants that reflect binding affinities using the BindingDB dataset. Our multi-agent LLM method significantly outperformed GPT-4 across multiple evaluation metrics by a significant margin. An ablation study revealed the contributions by each agent; ranked in terms of a contribution: the AI agent (i.e., ML prediction) was the most important followed by the KG agent then the search agent. The large contribution by the AI agent highlights the importance of LLM tool use in addressing questions that may not be part of text corpora. While our use case was related to biology, our presented architecture is applicable to other integrative prediction tasks. Code is available `https://anonymous.4open.science/r/DrugAgent-2BB7/`.

## 1 Introduction

Large Language Models (LLMs) have demonstrated remarkable capabilities in solving a wide range of problems using human-friendly inputs (Wei et al., 2022). However, these models still face limitations when confronted with tasks outside their training scope or those requiring real-time data access and specialized domain knowledge. To address these challenges, there is a growing interest in Multi-Agent systems (Du et al., 2023) that incorporate external tools, Knowledge Graphs (KG) (Shu et al., 2024), and Retrieval-Augmented Generation (RAG) (Lewis et al., 2020). These systems offer a more robust and reliable approach to problem-solving by leveraging diverse information sources and specialized capabilities. In this paper, we propose a novel multi-perspective, multi-agent system that integrates unstructured text, structured knowledge graphs, and machine learning predictions. This approach is designed to overcome the limitations of single-model systems and provide a more comprehensive solution to complex problems such as biomedical domain.

Our research focuses on two areas of drug discovery: drug-target interaction (DTI) prediction and drug repurposing. DTI prediction is a strategy for identifying potential targets for new compounds, which has the potential to significantly reduce the time, cost, and risk associated with drug development (Abbasi Mesrabadi et al., 2023). Drug repurposing, on the other hand, aims to identify new therapeutic uses for existing drugs, offering a faster path to clinical applications.

Pharmaceutical research is challenged by high failure rates due to the complexity of biological systems and the diversity of biomedical information sources (Chen et al., 2024; Wu et al., 2022b).

These challenges create a pressing need for innovative computational strategies that can effectively integrate and analyze vast, heterogeneous datasets (Huang et al., 2021; Lu, 2018). Recent advances in artificial intelligence, particularly in machine learning and knowledge graphs (Gyori et al., 2017), have helped address these challenges (Vamathevan et al., 2019). However, the effective integration of heterogeneous data sources and the interpretation of their complex interrelations remains a major research area.

To overcome these obstacles, we propose a multi-agent system framework where each agent specializes in a specific aspect of the drug discovery process. Our framework includes three primary agents; the AI agent, the KG Agent, and the Search Agent. The AI agent employs the DeepPurpose package (Huang et al., 2020) to predict DTIs. The KG Agent utilizes the drug-gene interaction database (DGIdb) (Cannon et al., 2024), DrugBank (Knox et al., 2024), Comparative Toxicogenomics Database (CTD) (Davis et al., 2023), and Search Tool for Interactions of Chemicals (STITCH) (Kuhn et al., 2007) to extract information on DTIs. The Search Agent engages with biomedical literature, LLMs for automated data labeling and validation. The AI Agent provides data-driven predictions, the KG Agent offers structured, curated knowledge, and the Search Agent contributes the latest findings. This multi-perspective approach enables a more comprehensive and accurate analysis of potential DTIs and drug repurposing opportunities.

The framework we developed, although initially designed for biological applications, can be adapted to various other fields requiring multi-perspective.

## 2 RELATED WORKS

The concept of drug target interaction has evolved with the advancements in computational tools, leading to a growing body of literature that explores various methodologies. Here, we highlight key developments in the field that align with our multi-agent system approach.

**Machine Learning in Drug Target Interaction and Repurposing**

Machine learning (ML) techniques have aided drug discovery, with use in various aspects of pharmaceutical research. Huang et al. (2020) is a deep learning models for DTI prediction that combines several algorithms (i.e., Graph Neural Networks (GNNs) and Convolutional Neural Networks (CNNs)). This model offers a versatile pre-trained approach applicable to a wide range of drug discovery tasks such as DTIs and drug property predictions. Similary in drug repurposing, Issa et al. (2021) demonstrating significant potential in predicting drug-disease interactions.

**Knowledge Graphs for Integrative Analysis.** Knowledge graphs provide a structured way of integrating diverse biological data. For instance, the DRKG, as employed by our Knowledge Graph Agent, integrates data from several sources, including DrugBank (Knox et al., 2024), Hetionet (Himmelstein et al., 2017), and STRING (Szklarczyk et al., 2023), to offer comprehensive insights into possible drug-disease links (Ioannidis et al., 2020). This structured integration facilitates the systematic exploration of DTI candidates.

**Literature Search using LLMs** The automation of literature review and data extraction using AI tools, particularly LLMs, has become a component of modern drug discovery (Chakraborty et al., 2023). Recent studies have demonstrated that LLM-based search tools can enhance the efficiency and complexity of queries compared to traditional search engines (Spatharioti et al., 2023). In our framework, we also leverage this approach by implementing a Search Agent that utilizes search engines as a data source.

**Multi-Agent Systems in Biomedical Applications.** While individual AI applications have shown promise, the integration of these technologies through a multi-agent system is less explored in the field of biomedical research though there are notable examples, such as clinical trials (Yue & Fu, 2024). These studies provide a foundation for the application of such multi-agent systems for drug discovery.

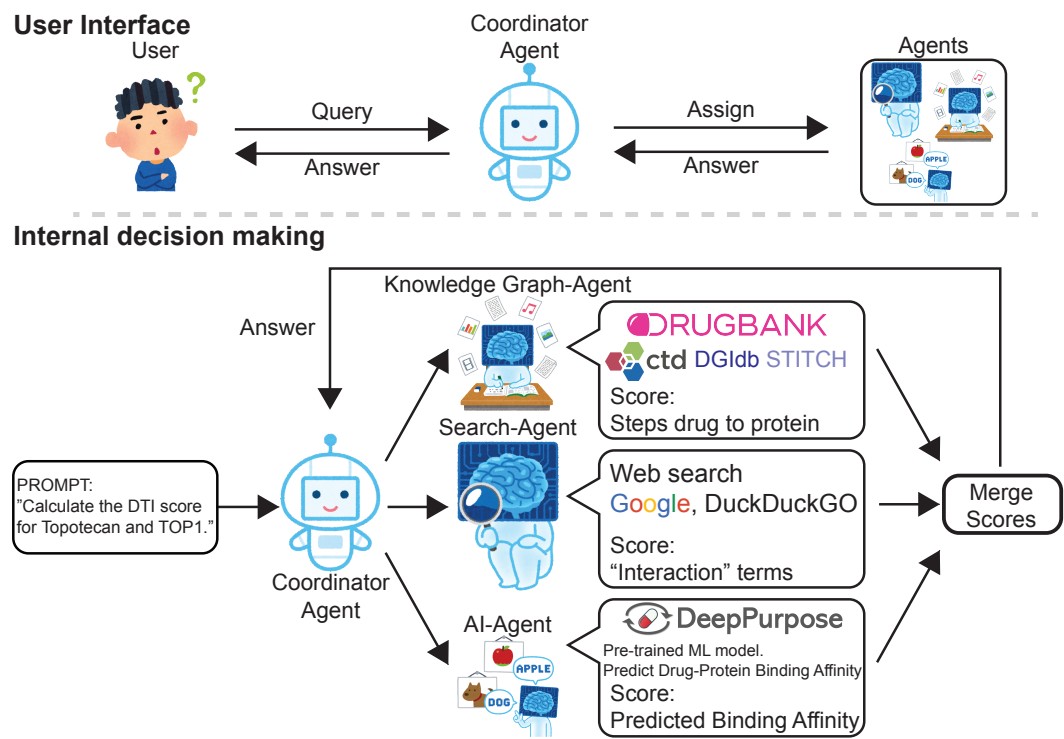

Figure 1: DrugAgent framework architecture for advanced DTI analysis. This system combines a user-friendly interface with sophisticated internal decision-making processes. It features a central "Coordinator" managing specialized agents: a "Knowledge Graph Agent" accessing biomedical databases (DrugBank, CTD, DGIdb, STITCH), a "Search Agent" utilizing web search engines, and an "AI Agent" employing deep learning models (trained on Davis, Kiba, BindingDB datasets with GNN, CNN, Transformers, etc). The system integrates RDKit and UniProt ID for chemical and protein data processing, culminating in a scoring function that synthesizes multi-source information to generate comprehensive answers for complex drug-target queries.

## 3 METHODS

Our DrugAgent framework is designed to mimic the collaborative and multidisciplinary nature of drug discovery teams with each agent in the system specialized to handle specific tasks. (See Appendix A.1 for detailed implementations of these agents.)

### 3.1 OVERVIEW OF DRUGAGENT

Our proposed system is a conversational multi-agent framework analogous to a specialized research team focused on drug target interaction prediction. Each agent within this system plays a distinct role, mirroring separate tasks research team members would do: some focus on machine learning models, others on search-based analysis, and another is dedicated to knowledge graph exploration.

The system comprises the following key agents:

1. A Coordinator Agent that oversees the specialized agents and integrates their findings.

2. An AI Agent specializing in predicting DTI potential using machine learning models.

3. A Search Agent focusing on analyzing existing literature and data for repurposing opportunities.

4. A Knowledge Graph (KG) Agent dedicated to exploring connections between drugs, diseases, and biological pathways.

The system employs LLMs for natural language processing and response generation, enhances reasoning through step-by-step problem-solving methodologies, and performs actions like calculating scores, analyzing literature, and querying knowledge graphs. It then integrates this information using a weighted average approach to simulate a knowledgeable DTI research team.

### 3.2 Agent Roles and Responsibilities

The DrugAgent framework integrates a diverse array of specialized agents, each employing the ReAct (Yao et al., 2022) and LEAST-TO-MOST (Zhou et al., 2022) reasoning methods to plan their actions. Through the use of advanced search capabilities, access to specialist models, and indexing in databases, these agents can execute a wide range of tasks effectively. Below, we delve into the specific roles and responsibilities assigned to each agent within the system.

### 3.3 AI Agent

Our approach begins with the AI Agent, which utilizes the MPNN_CNN_BindingDB model from DeepPurpose (Huang et al., 2020) to predict potential drug-target. This model combines Message Passing Neural Networks (MPNN) (Gilmer et al., 2017) for processing molecular structures with CNN for embedding binding site features. It is trained on the comprehensive BindingDB dataset, which contains binding affinity data for DTIs. DeepPurpose can predict binding affinity values for any combination of SMILES and target sequence provided.

The MPNN_CNN_BindingDB model operates as follows:

1. The MPNN component processes the molecular graph of the drug, capturing its structural features.
2. The CNN component analyzes the binding site information of the target protein.
3. These features are then combined and processed through fully connected layers to predict binding affinity.

### 3.4 Knowledge Graph (KG) Agent

Concurrently, the Knowledge Graph (KG) Agent employs DGIdb (Cannon et al., 2024), DrugBank (Knox et al., 2024), CTD (Davis et al., 2023), and STITCH (Kuhn et al., 2007). From these datasets, we make use of the DTI table and then create the drug-gene interaction table. This consolidated table contains 3,312 drugs and 23,066 genes. From this, we calculate the number of hops to reach from the drug to the target using the below formula,

$$\text{DTI}_{\text{score}}(d, t) = \begin{cases} 0 & \text{if } d \notin G \text{ or } t \notin G, \\ 1 & \text{if } h(d, t) = 1, \\ \frac{1}{\ln(1+h(d,t))} & \text{otherwise,} \end{cases} \tag{1}$$

where $d$ is a drug, $t$ is a target $G$ is a knowledge graph $h(d, t)$ is a number of hops in the shortest path between $d$ and $t$ in $G$ and $\ln(\cdot)$ is a natural logarithm. The score decreases logarithmically as the path length between drug and target increases.

### 3.5 Search Agent: Information Extraction from Biomedical Literature

Parallel to these processes, the Search Agent leverages LLMs to automate the extraction of relevant information from biomedical literature found via search engine hits. This agent applies natural language processing techniques to extract and annotate data regarding drug efficacy and novel interactions, which are critical for validating and updating the predictions generated by the other agents.

The search agent's core functionality can be summarized as follows:

1. **Google Search Query**: The agent formulates a search query combining the drug name and target name, along with the term "interaction".
2. **Web Scraping**: It performs a Google search using this query and scrapes the search results, including titles, links, and snippets.

3. **Text Analysis**: The agent analyzes the scraped text for the presence of the drug name, target name, and predefined keywords related to interactions and efficacy.

4. **Scoring**: Based on the presence of these elements, it assigns a score to each search result. The scoring system considers: (1) Presence of both drug and target names; (2) Occurrence of interaction-related keywords; (3) The presence of words indicating strong or significant effects.

5. **DTI Score Calculation**: Finally, it calculates an overall DTI score by aggregating individual result scores and normalizing the total.

This simplified approach allows for rapid information gathering from publicly available sources. However, it is important to note that this method relies on the quality and relevance of Google search results, and does not analyze full scientific papers or curated databases. As such, it serves as a preliminary screening tool rather than a comprehensive literature review system.

The DTI score calculation is as follows: Let $R = \{r_1, r_2, ..., r_n\}$ be the set of search results, where $n$ is the number of results (default $n$ is 10). For each result $r_i$, we define an individual score function $S(r_i)$: $S(r_i) = I(d, t, r_i) + I(p, r_i) + I(s, r_i)$, where

$$I(d, t, r_i) = \begin{cases} 1 & \text{if drug name } d \text{ and target name } t \text{ are in } r_i \\ 0 & \text{otherwise,} \end{cases}$$

$$I(p, r_i) = \begin{cases} 1 & \text{if any positive keyword is in } r_i \\ 0 & \text{otherwise,} \end{cases}$$

and

$$I(s, r_i) = \begin{cases} 1 & \text{if any strong keyword is in } r_i \\ 0 & \text{otherwise.} \end{cases}$$

The positive keywords are "interacts", "binds", "activates", "inhibits", and "modulates". The strong keywords are "strong", "significant", "potent", and "effective".

The total score $T$ is then calculated as $T = \sum_{i=1}^{n} S(r_i)$. The maximum possible score $M$ is $M = 3n$. Finally, the normalized DTI score $D$ is calculated as:

$$D = \begin{cases} \text{round}\left(\frac{T}{M}, 2\right) & \text{if } M > 0 \\ 0 & \text{if } M = 0, \end{cases} \tag{2}$$

where $\text{round}(x, 2)$ rounds $x$ to 2 decimal places.

### 3.6 CALLING EXTERNAL TOOLS

OpenAI GPT supports calling external tools (e.g., function, database retrieval) to leverage external knowledge and enhance its capability. Specifically, suppose we have multiple tools, GPT's API can detect which tool to use, which serves as glue to connect LLMs to external tools. Our system integrates external data sources and predictive AI models. (See AppendixA.3 for detailed external tool information.)

**Data Sources**  The use of professional datasets is pivotal in ensuring the accuracy and reliability of our agents' information retrieval capabilities.

- **DrugBank:** DrugBank (Knox et al., 2024) offers detailed drug data, including chemical, pharmacological, and pharmaceutical information, with a focus on comprehensive DTIs. It provides data for over 13,000 drug entries, including FDA-approved small-molecule drugs, FDA-approved biopharmaceuticals (proteins, peptides, vaccines, and allergens), and nutraceuticals.

- **Comparative Toxicogenomics Database (CTD):** The CTD (Davis et al., 2023) is a curated database that provides information about chemical–gene/protein interactions, chemical–disease, and gene-disease relationships. It is valuable for understanding how environmental exposures affect human health, integrating data from various species and linking chemicals, genes, diseases, phenotypes, and pathways (Chang et al., 2019; Wu et al., 2022a).

- **Search Tool for Interactions of Chemicals (STITCH):** STITCH (Kuhn et al., 2007) is a database of known and predicted interactions between chemicals and proteins. It integrates information from various sources, including experimental data, predictive methods, and text-mining of scientific literature. STITCH is useful for exploring the complex network of interactions between drugs, other chemicals, and proteins.

- **Drug-Gene Interaction Database (DGIdb):** DGIdb (Cannon et al., 2024) is a resource that consolidates disparate data sources describing drug-gene interactions and gene druggability. It provides drug-target interaction and information on druggable genes used in cancer informatics, drug repurposing, and personalized medicine (Chen et al., 2021; Wang et al., 2024; Lu et al., 2024).

**Predictive AI Models** We utilize DeepPurpose (Huang et al., 2020) for the AI Agent. DeepPurpose is a comprehensive and extensible deep learning library for DTI prediction. It integrates multiple state-of-the-art models and datasets, allowing researchers to implement various deep learning approaches for drug discovery and repurposing. DeepPurpose facilitates the application of AI in drug development by providing a unified framework for different drug and protein encoding methods.

### 3.7 SCORING FUNCTION USED BY COORDINATOR AGENT

The Coordinator Agent utilizes a scoring function to synthesize predictions from the AI, KG, and Search Agents. This function cross-references and enriches AI agent predictions with data from the KG Agent and validates them against findings from the Search Agent. This integrated approach enables a dynamic updating mechanism, where feedback from literature and knowledge graph analyses continually refines the predictions.

The scoring function calculates the final prediction score using this formula:

$$S_{merged} = \alpha S_{AI} + \beta S_{KG} + \gamma S_{Search},$$

where $S_{merged}$ is the merged DTI score, $S_{AI}$ is the AI-based DTI score, $S_{KG}$ is the knowledge graph-based DTI score, $S_{Search}$ is the search-based DTI score, $\alpha$, $\beta$, and $\gamma$ are the weights assigned to the AI, knowledge graph, and search-based scores, respectively. (See AppendixA.2 for detailed implementations of weight optimization.)

### 3.8 WORKFLOW

The workflow of our DrugAgent is designed to leverage the strengths of multiple specialized agents to provide comprehensive and accurate DTI scores. The process is structured in several sequential steps, as described below:

**Step 1: Query Initialization and Agent Preparation.** The workflow begins with the user input, specifying the drug name, target name, and weighting parameters ($\alpha$, $\beta$, and $\gamma$). The system initializes four specialized agents: the AI Agent, Search Agent, Knowledge Graph (KG) Agent, and Coordinator Agent.

**Step 2: Task Allocation to Specialist Agents.** The Coordinator Agent, acting as the central manager, allocates specific tasks to each specialist agent:

- The AI Agent is tasked with calculating the DTI score using machine learning models.

- The Search Agent is responsible for analyzing DTI data using search methods and literature analysis.

- The KG Agent is assigned to analyze DTI data using Knowledge Graph techniques.

**Step 3: Independent Agent Processing.** Each specialist agent processes its assigned task independently, utilizing its specific methodologies and tools:

- The AI Agent applies machine learning models to predict the DTI score.

- The Search Agent conducts literature searches and analyzes the results to derive a DTI score.

- The KG Agent queries and analyzes the knowledge graph to determine the DTI score.

Table 1: Performance comparison between DrugAgent and GPT-4. We report the average results of 5 independent runs and the corresponding standard deviations (in brackets). For each metric, we highlight the best method in **bold**. We marked the metrics where DrugAgent is better than GPT-4 (pass the t-test, i.e., p-value<0.05) using "*".

| Metric | DrugAgent | GPT-4 | w.o. AI Agent | w.o. KG Agent | w.o. Search Agent |
|---|---|---|---|---|---|
| MSE ($\downarrow$) | **1.836*** | 13.420 | 52.349 | 8.119 | 1.960 |
| | (0.007) | (0.042) | (0.051) | (0.023) | (0.000) |
| MAPE ($\downarrow$) | **0.134*** | 0.320 | 0.808 | 0.312 | 0.138 |
| | (0.000) | (0.000) | (0.001) | (0.001) | (0.000) |
| MAE ($\downarrow$) | **1.081*** | 3.350 | 7.095 | 2.706 | 1.124 |
| | (0.003) | (0.000) | (0.005) | (0.005) | (0.000) |
| R2 ($\uparrow$) | **0.431*** | -0.460 | -15.228 | -1.517 | 0.393 |
| | (0.002) | (0.001) | (0.016) | (0.007) | (0.000) |
| Explained Variance ($\uparrow$) | **0.577*** | -0.460 | 0.378 | 0.211 | 0.572 |
| | (0.003) | (0.001) | (0.006) | (0.002) | (0.000) |
| Max Error ($\downarrow$) | **2.809*** | 6.490 | 9.639 | 4.395 | 2.902 |
| | (0.014) | (0.120) | (0.006) | (0.014) | (0.000) |
| Correlation ($\uparrow$) | **0.761*** | 0.110 | 0.708 | 0.507 | 0.758 |
| | (0.002) | (0.003) | (0.011) | (0.001) | (0.000) |
| Runtime ($\downarrow$) | $\approx$5.000s | $\approx$0.297s | - | - | - |
| # OpenAI API tokens ($\downarrow$) | $\approx$2000-3000 | $\approx$100 | - | - | - |
| cost of tokens ($\downarrow$) | $\approx$\$0.006-\$0.027 | $\approx$\$0.0014-\$0.0020 | - | - | - |

**Step 4: Score Collection and Merging.** After each agent completes its task, the individual DTI scores are reported back to the Coordinator Agent. The Coordinator merges these scores, applying the provided weighting parameters ($\alpha$, $\beta$, and $\gamma$) to the individual scores for a final DTI score.

**Step 5: Result Integration and Final Output.** The Coordinator Agent integrates all the information, including the individual scores from each method and produces a merged final score. It formats this information into a structured output, providing a comprehensive view of the DTI prediction from multiple perspectives.

**Step 6: Delivery of Solution.** The final output, which includes the merged DTI score along with the individual scores from each method, is delivered to the user. This comprehensive result provides not only the final prediction but also insights into how different methods contribute to the overall score, enhancing the user's understanding of the DTI prediction. This structured workflow ensures that our multi-agent DTI prediction system combines multiple analytical approaches, offering a robust and multi-faceted assessment of potential DTIs.

## 4 EXPERIMENT

In this section, we demonstrate the experimental results and case studies. Due to the page limit, the experimental setups, including dataset description, evaluation metrics, and implementation details, are elaborated in the Appendix B.

### 4.1 QUANTITATIVE RESULTS: PERFORMANCE COMPARISON OF DRUGAGENT AND GPT-4

To evaluate the performance of DrugAgent against GPT-4, we predict pKd scores and compere these to values in BindingDB database using several key statistical metrics. pKd is a measure of the binding affinity between a drug and its target protein, expressed as the negative logarithm of the dissociation constant (Kd). Higher pKd values indicate stronger binding affinity. Table 1 summarizes the results for 10 diverse drug-target combinations not used in parameter tuning.

DrugAgent outperformed GPT-4 across all examined metrics. Regarding prediction accuracy, DrugAgent achieved a Mean Squared Error (MSE) of 1.836, lower than GPT-4's 13.420, indicating superior overall predictive power. The Mean Absolute Percentage Error (MAPE) and Mean Absolute Error (MAE) metrics further confirm this superiority, with DrugAgent achieving values of 0.134 and 1.081, respectively, compared to GPT-4's 0.320 and 3.350.

DrugAgent demonstrated stronger explanatory power, evidenced by its positive R-squared (R2) value of 0.431, indicating a moderate fit to the data. In contrast, GPT-4's R2 value of -0.460 suggests a poor fit. The Explained Variance metric reinforces this trend, with DrugAgent achieving a positive

value of 0.577, while GPT-4 showed a value of -0.460. These results highlight DrugAgent's superior ability to capture and explain the variance in the pKd score data.

DrugAgent also excels in prediction consistency. Its Max Error of 2.809 is less than half of GPT-4's 6.490, indicating more reliable predictions across the dataset. Moreover, the strong positive correlation (0.761) between DrugAgent's predicted and actual values, compared to GPT-4's weak positive correlation (0.110), underscores DrugAgent's effectiveness in capturing the underlying relationships in the data.

These results showcase DrugAgent's improved predictive capabilities for BindingDB pKd scores compared to GPT-4, which is crucial for accurate binding affinity predictions in drug discovery and molecular interaction studies.

Our ablation study provides valuable insights into the importance of each component in the DrugAgent architecture. Removing the AI Agent resulted in the most severe performance degradation across all metrics, with MSE increasing to 52.349 and R2 declining to -15.228. This underscores the AI Agent's critical role in understanding complex patterns in molecular structures and their relationship to binding affinities. The KG Agent also proved essential, as its removal led to significant performance drops, though less severe than the AI Agent. This indicates the KG Agent's crucial contribution of domain knowledge about chemical structures and known interactions. While the Search Agent had a less dramatic impact on performance, it still contributed to the overall accuracy of the model, particularly in maintaining high correlation and explained variance.

DrugAgent achieves its superior performance with a runtime of 5.000s, which is efficient considering the complexity of its multi-agent architecture. In comparison, GPT-4's runtime of 0.297s is faster, but at the cost of reduced accuracy.

We also compared the number of OpenAI API tokens used and the associated costs. DrugAgent uses between 2000-3000 tokens per prediction, with an approximate cost of \$0.006-\$0.027, while GPT-4 uses around 100 tokens, costing $0.0014-0.0020$ per prediction. While DrugAgent has higher token usage and cost, its superior performance justifies this increased resource utilization for applications requiring high accuracy.

This comprehensive analysis highlights the contributions between the AI, KG, and Search Agents in our model architecture for accurate pKd score prediction. The work also provides valuable insights for future improvements of DrugAgent. They emphasize the potential for enhancing the integration and capabilities of each component to achieve better score predictions in the context of the BindingDB database, while considering the balance between performance, computational resources, and cost.

## 4.2 CASE STUDY

This study presents three case studies analyzing the drug target potential of Topotecan for different proteins: TOP1, SLFN11, and SLC26A4. These cases represent a spectrum from known strong interactions to potentially novel connections, allowing for an evaluation of our multi-agent system's capabilities.

Case 1 examined the interaction between Topotecan and TOP1, a known strong drug-target interaction. The system calculated a final score of 11.51, confirming the established relationship. The high AI Agent score (7.65) and KG Agent score (1.0) aligned with the known mechanism of Topotecan as a TOP1 inhibitor, while the relatively low Search Agent score (0.27) reflected the well-established nature of this interaction not requiring extensive new studies. The high scores from the AI and KG agents further support the strong interaction between Topotecan and TOP1.

Case 2 investigated the less understood but potentially relevant interaction between Topotecan and SLFN11. The system yielded a final score of 10.30, suggesting a noteworthy relationship. The high AI Agent score (7.36) indicated structural compatibility, while the moderate Search Agent (0.33) and KG Agent (0.72) scores reflected some existing evidence and established connections, albeit less comprehensive than in Case 1.

Case 3 explored an unlikely interaction between Topotecan and SLC26A4, resulting in a final score of 9.92. Despite a high AI Agent score (7.61) suggesting structural compatibility, the very low Search Agent score (0.00) indicated minimal literature evidence. The moderate KG Agent score (0.72) suggested some indirect connections, highlighting the system's ability to detect potential novel

interactions. However, due to the lack of direct evidence and the complexity of these interactions, careful interpretation is required.

These case studies demonstrated the multi-agent system's ability to handle several scenarios related to DTI. The system integrated various data sources and analytical methods, providing interpretable results with detailed reasoning processes.

The system offered practical insights for DTI research across different levels of prior knowledge. In Case 1, it confirmed a well-established interaction. In Case 2, it suggested a potentially relevant interaction that warrants further investigation. In Case 3, it identified a possible novel interaction while flagging the low literature evidence, thus highlighting an area requiring careful experimental validation.

## 5    DISCUSSION

Our study presents a novel multi-agent system for DTI prediction and drug repurposing that integrates machine learning, knowledge graphs, and literature search. This approach offers more robust predictions by leveraging diverse data sources and analytical methods. The system's strength lies in its collaborative approach, which combines each agent's specialized capabilities to evaluate complex DTIs. The weighted integration method allows for flexible adjustment of different prediction methods, enhancing overall accuracy.

However, limitations exist. The system still relies on human expertise for initial setup, limiting its scalability. It also lacks autonomous knowledge updating capabilities to keep pace with evolving pharmacological research and requires regular database updates to maintain its relevance and accuracy. Furthermore, the current model would need to be paired with a separate system to make use of individual patient characteristics.

The system's applicability to a wider range of research tasks can be done by incorporating more existing models is essential. This flexibility will facilitate easy adaptation to various drug tasks without major architectural changes, such as drug synergy prediction (Huang et al., 2022), drug property prediction (Xu et al., 2024), drug response prediction (Inoue et al., 2024b), adverse drug reaction prediction (Chen et al., 2024), and drug design (Fu et al., 2021a; 2022).

Multi-agent systems can be effectively applied to automate the preprocessing of other data, such as single-cell RNA sequencing (scRNA-seq) data. This involves implementing multiple preprocessing functions including imputation (Inoue et al., 2024a; Inoue, 2024), quality control (McCarthy et al., 2017; Lu et al., 2023), and batch effect correction (Li et al., 2020; Fu et al., 2024; Haghverdi et al., 2018). Developing an optimization framework to automatically select and apply the most appropriate preprocessing methods for given datasets, and integrating these preprocessing capabilities into the existing multi-agent system, will expand system utility.

In conclusion, our DrugAgent shows promise in accelerating AI-driven drug discovery. Continued development addressing both computational and pharmacological challenges could lead to more efficient and cost-effective drug discovery processes(Zhang et al., 2021). Future work should focus on validating the system in real-world drug discovery projects (Fu et al., 2021b) and evaluating its performance with larger, more diverse datasets.

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

## A IMPLEMENTATION

### A.1 IMPLEMENTATION DETAILS

In this section, we provide detailed descriptions of the implementation processes to enhance the reproducibility of our study.

**Role Assignment to Agents**    Each agent within our multi-agent framework is designated a specific role, which is integrated directly into the LLM's system prompt for clarity and focus. For instance, the role of the AI Agent is defined as follows:

> """Specialized AI Agent for calculating DTI scores using machine learning models.
> Use the get_AI_score function to obtain the DTI score. Output the score in the
> following format: { "AI_score": 1.0,}"""

This role definition is crucial as it guides the LLM to prioritize responses based on the assigned expert domain, leveraging the model's inherent capability to focus more acutely on instructed tasks than on general information.

**Defining External Tools**    External tools are defined in a structured format to facilitate their integration and usage within the LLM environment. These definitions are crafted in Python functions, specifying the function name, parameters, and return types. Key examples include:

1. AI Agent: Utilizes machine learning models for DTI scoring.
2. Search Agent: Performs web-based information retrieval to gather relevant DTI data.
3. KG Agent: Leverages a knowledge graph (KG) for graph-based DTI scoring.

This structured approach allows for the direct execution of function calls within the system, providing detailed responses, including the function name and arguments. These responses enable the retrieval of results in a structured manner.

**Enhanced Score Integration**    To improve the model's prediction capabilities, we incorporate a weighted integration method within the Coordinator Agent. This method, known as score merging, aids in synthesizing the outputs from different agents into a comprehensive DTI prediction. The integration is performed using predefined weights ($\alpha$, $\beta$, and $\gamma$) to balance the contributions of each prediction method:

$$\text{merged\_dti\_score} = \alpha * \text{AI\_score} + \beta * \text{KG\_score} + \gamma * \text{search\_score}.$$

Here, `AI_score`, `search_score`, and `KG_score` represent the DTI scores from the AI Agent, Search Agent, and KG Agent, respectively. The weights $\alpha$, $\beta$ $\gamma$ are adjustable hyperparameters that determine the relative importance of each score in the final prediction. This approach enhances the accuracy of the model's outputs and its ability to leverage diverse prediction methods for more robust drug repurposing predictions. This parameter and the formula were defined systematically. (See A.2 in detail.)

These implementation strategies collectively ensure that each component of our multi-agent system operates effectively and that the integration between different agents and external tools is seamless, fostering an environment conducive to robust, reproducible research in drug repurposing prediction.

**Software and Hardware Configuration**    Our experimental framework was implemented on a Mac computer equipped with an Apple M1 chip and 16GB unified memory, utilizing the built-in GPU cores. We used Python 3.10 for scripting, PyAutoGen 0.2.31 (Wu et al., 2023), DeepPurpose 0.1.5 (Huang et al., 2020), and RDKit 2023.9.6 (Landrum et al., 2024). For each experiment, we used the same seed to ensure reproducibility across different Mac models.

### A.2 PATTERN SELECTION AND WEIGHT OPTIMIZATION FOR SCORE INTEGRATION

To determine the optimal integration method for our agent scores, we explored four different mathematical patterns and employed a constrained optimization method. We utilized a dataset comprising

3,332 drug-target pairs, each containing scores from our three specialized agents (AI, Knowledge Graph, and Search) along with corresponding ground truth interaction scores (pKd values) from the BindingDB dataset. We considered the following four patterns for score integration:

$$
\begin{aligned}
f_1(\alpha, \beta, \gamma, A, B, C) &= \alpha A + \beta B + \gamma C, \\
f_2(\alpha, \beta, \gamma, A, B, C) &= \alpha A + (\beta B \cdot \gamma C), \\
f_3(\alpha, \beta, \gamma, A, B, C) &= (\alpha A \cdot \beta B) + \gamma C, \\
f_4(\alpha, \beta, \gamma, A, B, C) &= (\alpha A \cdot \gamma C) + \beta B,
\end{aligned}
\tag{3}
$$

where $A$, $B$, and $C$ represent the scores from the AI, Knowledge Graph, and Search agents respectively, and $\alpha$, $\beta$, and $\gamma$ are the weights we are optimizing. For each pattern, we formulated the weight optimization as a constrained minimization problem:

$$
\begin{aligned}
&\underset{\alpha, \beta, \gamma}{\text{minimize}} \quad \|Y - f(\alpha, \beta, \gamma, A, B, C)\|_2 \\
&\text{subject to} \quad \alpha, \beta, \gamma \geq 0,
\end{aligned}
\tag{4}
$$

where $Y$ is the vector of ground truth pKd values and $i \in 1, 2, 3, 4$ corresponds to the pattern type. The optimization was performed using Sequential Least Squares Programming (SLSQP) with non-negative constraints for each pattern. After comparing the results, we found that the linear combination pattern $f_1$ yielded the best performance:

$$
f_1(\alpha, \beta, \gamma, A, B, C) = \alpha A + \beta B + \gamma C.
\tag{5}
$$

The initial optimized weights for this pattern were:

$$
\begin{aligned}
\alpha &= 1.24683589 \\
\beta &= 2.23513134 \\
\gamma &= 3.22163745 \times 10^{-16}
\end{aligned}
\tag{6}
$$

After obtaining these initial results, we rounded and adjusted the coefficients for practical implementation and to account for potential overfitting to our specific dataset. The final weights used in our merged DTI score calculation are:

$$
S_{\text{merged}} = 1.2 S_{\text{AI}} + 2.2 S_{\text{KG}} + 0.5 S_{\text{Search}}.
\tag{7}
$$

While initial optimization suggested a negligible contribution from the Search agent, we assigned it a small but non-trivial weight of 0.5. This decision maintains model flexibility and acknowledges the potential value of diverse information sources in future applications or different datasets, even if not significantly impactful in our current study.

The selection of the linear combination pattern ($f_1$) and the subsequent weight adjustments reflect several important considerations:

The linear pattern provided the best fit to our data while maintaining simplicity and interpretability. The Knowledge Graph (KG) agent retains the highest weight, confirming its significant contribution to the final prediction. The AI agent continues to play a substantial role, with a weight slightly lower than the KG agent. While the initial optimization suggested a negligible role for the Search agent, we maintained its contribution at a non-trivial level (0.5) in the final model. This adjustment reflects our belief in the potential value of diverse information sources, even if not prominently represented in our current dataset.

It is worth noting that the sum of these weights (3.9) is intentionally not normalized to 1. This allows for a more flexible scaling of the final score, which can be beneficial in certain applications or when comparing across different datasets. This approach, combining pattern selection, data-driven weight optimization, and expert adjustment, ensures that our final predictions leverage the strengths of each agent while maintaining robustness and generalizability. The results suggest that a linear combination of structured knowledge from the KG agent, pattern recognition capabilities of the AI agent, and supplementary information from the Search agent contributes to accurate pKd value prediction, with the KG and AI agents playing particularly crucial roles.

## A.3 DETAILED EXTERNAL TOOL DEFINITIONS

This appendix provides a comprehensive overview of the implementation details for our three key agents: AI Agent, Search Agent, and KG Agent. Each agent plays a crucial role in our multi-agent system for drug repurposing prediction.

### A.3.1 AI AGENT IMPLEMENTATION

The AI Agent utilizes machine learning models to predict DTIs. Its core function, get_AI_score, takes a drug name and a target name as input and returns a float value representing the predicted interaction score.

```
# AI Agent
def get_ml_dti_score(name: str, target_name: str) -> float:
    target_sequence = get_target_sequence(target_name)
    net = models.model_pretrained(model="MPNN_CNN_BindingDB")

    X_repurpose, drug_name, drug_cid = load_broad_repurposing_hub(
        SAVE_PATH
    )
    idx = drug_name == name
    if not any(idx):
        print(f"Logging: Drug '{name}' not found.")
        return None

    res = models.virtual_screening(
        X_repurpose[idx], [target_sequence], net,
        drug_name[idx], [target_name]
    )
    return res[0]
```

This implementation uses a pre-trained MPNN_CNN model from the BindingDB dataset. It first retrieves the target protein sequence and loads the drug data. If the specified drug is found, it performs virtual screening to predict the interaction score.

### A.3.2 SEARCH AGENT IMPLEMENTATION

The Search Agent leverages web-based information to gather relevant data about DTIs. It consists of several functions that work together to perform a Google search, parse the results, and calculate a DTI score based on the search findings.

```
# Search Agent
def google_search(query: str, num_results: int = 10) -> List[Dict[str, str]]:
    # ... [implementation details]

def _parse_search_results(soup: BeautifulSoup) -> List[Dict[str, str]]:
    # ... [implementation details]

def calculate_dti_score(search_results: List[Dict[str, str]],
                        drug_name: str, target_name: str) -> float:
    # ... [implementation details]

def _calculate_individual_score(result: Dict[str, str], drug_name: str,
                                target_name: str, positive_keywords: List[str],
                                strong_keywords: List[str]) -> int:
    # ... [implementation details]

def analyze_dti(name: str, target_name: str) -> float:
    search_results = google_search(f"{name} {target_name} interaction")
    dti_score = calculate_dti_score(search_results, name, target_name)
    return dti_score
```

The main function, analyze_dti, orchestrates the search process and score calculation. It uses a keyword-based scoring system to evaluate the relevance and strength of the interaction based on search results.

### A.3.3 KG AGENT IMPLEMENTATION

The KG Agent utilizes a knowledge graph to derive DTI scores based on the structural relationships between drugs and targets in the graph.

```
# KG Agent
def calculate_dti_score(kg, drug, target):
    if drug not in kg.graph or target not in kg.graph:
        return 0  # Return 0 if the drug or target is not in the knowledge graph

    hops = kg.shortest_path(drug, target)
    if hops == -1:
        return 0  # No relationship
    elif hops == 1:
        return 1  # Direct connection
    else:
        return 1 / (np.log1p(hops))  # Logarithm-based score

def load_kg(file_path):
    with open(file_path, "rb") as f:
        kg = pickle.load(f)
    return kg

def get_kg_dti_score(name: str, target_name: str) -> float:
    kg = load_kg("../data/knowledge_graph.pkl")
    score = calculate_dti_score(kg, name, target_name)
    return score
```

The KG Agent loads a pre-constructed knowledge graph and calculates the DTI score based on the shortest path between the drug and target nodes in the graph. A direct connection yields the highest score, while more distant connections result in lower scores, calculated using a logarithmic scale.

These detailed implementations demonstrate how each agent contributes unique insights to the overall DTI prediction task. The AI Agent provides predictions based on learned patterns from large datasets, the Search Agent incorporates up-to-date information from web sources, and the KG Agent leverages structured knowledge representations. By combining these diverse approaches, our system aims to produce more robust and comprehensive drug repurposing predictions.

## B EXPERIMENTAL SETUP

### B.1 DATASET AND EVALUATION METRICS

In our study, we utilized the Kd (dissociation constant) data from the BindingDB database (Liu et al., 2007) as our experimental dataset. BindingDB is a public repository of measured binding affinities, primarily focusing on interactions between proteins considered as drug targets and small, drug-like molecules.

The Kd dataset comprises 52,284 DTI pairs, involving 10,665 unique drug-like compounds and 1,413 distinct protein targets. Kd values represent the dissociation constant, which quantifies the propensity of a larger complex to separate (dissociate) into smaller components. A lower Kd value indicates a higher binding affinity between the drug and the target protein.

Our regression task involved predicting pKd (negative logarithm of the dissociation constant Kd) values from the BindingDB database, given the target protein's amino acid sequence and the drug compound's SMILES string (Simplified Molecular-Input Line-Entry System, a line notation for encoding molecular structures) (Weininger, 1988). This task is crucial for understanding Drug-Target Interactions (DTIs) and has significant implications for drug discovery and development processes (Huang et al., 2022).

To comprehensively evaluate our model's performance, we employed a suite of seven complementary metrics: Mean Squared Error (MSE), Mean Absolute Percentage Error (MAPE), Mean Absolute Error (MAE), R-squared (R2) Score, Explained Variance, Maximum Error, and Correlation. These metrics collectively assess various aspects of our predictions: MSE and MAE provide measures of the average prediction error, with MSE being more sensitive to large errors due to its quadratic nature. MAPE offers insight into the relative size of prediction errors. The R2 score and Explained Variance evaluate the model's capacity to capture the underlying variance in the data. Maximum Error highlights the worst-case prediction scenario, crucial for understanding the model's limitations.

Lastly, Correlation assesses the strength and direction of the relationship between predicted and actual pKd values.

This comprehensive set of metrics allows us to thoroughly assess the accuracy of our predictions, the model's capacity to capture underlying patterns in DTIs, and its consistency across different scenarios. Such a multi-faceted evaluation is essential for validating the model's performance and identifying areas for potential improvement in the context of pKd score prediction for drug-target interactions. Due to space limitation, implementation details are provided in Appendix (Section A.1).

### B.2 BASELINE SETUP

This section describes the baseline setup for predicting pKd values for drug-target interactions using GPT-4. First, the necessary environment setup is performed to use the OpenAI API. The Python client for OpenAI is imported, and the API key is retrieved from an environment variable. If the API key is not set, an error is raised. Next, a list of drug-target combinations for prediction is defined. This list includes drugs such as Gefitinib, Sumatriptan, and Betaxolol, along with their corresponding target proteins (e.g., PRKACB, KDR, HTR2C). After initializing the OpenAI client, the code loops through each drug-target combination. Within the loop, a prompt is sent to the GPT-4 model to predict the pKd value for the specific drug and target. The request to GPT-4 includes two messages:

A system message: This instructs the AI to take on the role of predicting pKd values for drug-target interactions and to return only a single numeric value. A user message: This requests a pKd value prediction for the specific drug and target combination.

After receiving the response from GPT-4, the code verifies that the returned value is a number. If valid, it outputs the value. Invalid responses (non-numeric) are handled as error messages. This setup provides a basic framework for using GPT-4 to predict pKd values for drug-target interactions. It allows for rapid predictions across a large set of drug-target combinations, potentially aiding in the screening of candidate drugs in the early stages of the drug discovery process.

### B.3 PROCEDURE

Each agent in our DrugAgent system was tasked with specific roles, as outlined in the Methods section. The AI Agent applied machine learning models to calculate the DTI score, the Search Agent analyzed literature data to derive a DTI score based on published research, and the KG Agent evaluated DTIs using graph-based techniques. The Coordinator Agent then synthesized these findings into a comprehensive DTI prediction. We conducted experiments to assess the accuracy of the merged DTI scores and the consistency of predictions across different methods.

