# OpenReview forum: "DrugAgent: Multi-Agent Large Language Model-Based Reasoning for Drug-Target Interaction Prediction and Repurposing"
_ICLR.cc/2025/Conference — Submitted to ICLR 2025_

### Official Review · Reviewer_emEv · 2024-11-02

**Soundness:** 1
**Presentation:** 1
**Contribution:** 1
**Rating:** 1
**Confidence:** 5

**Summary:**

The manuscript entitled “DrugAgent: Multi-Agent Large Language Model-Based Reasoning for Drug-Target Interaction Prediction and Repurposing” propose a multi-agent system framework, named DrugAgent, that focus on drug-target interaction prediction and drug repurposing. Specifically, the proposed DrugAgent designs three types of agent to integrate unstructured text, structured knowledge graphs, and machine learning predictions, respectively.

**Strengths:**

The main contribution of this paper is an architecture exploiting multi-agents LLM to detect drug-target interaction and drug repurposing. The system is composed by the AI agent, the KG Agent, and the Search Agent, to predict interaction between drug-target pairs. The first uses DeepPurpose as encoder to obtain drug representation, while the second agent is used to extract knowledge on DTIs, followed by a search agent for automated data labeling and validation.

**Weaknesses:**

1.The contribution of this paper is limited. The proposed model mostly reuses well-known techniques from literature (e.g., DeepPurpose), integrated in a seemingly correct (and mostly non-innovative way).

2.The idea is reasonable and convincing, which seems it focuses more on engineering. The implementation is sensible but does not add much to existing body-of-work in ML/AI. We would have expected to see some more elaborations as concerns the proposed technique.

3.The writing skill is not mature enough, and the relationship between multi-agents system and downstream tasks is not clear. It can also be established by changing to other related tasks such as DDI prediction.

4.The experimental results is not enough to demonstrate the performance of proposed method. As a key problem in drug discovery, many state-of-the-art deep learning methods are proposed to predict DTIs.

**Questions:**

Please see my comments for weaknesses.

---

### Official Review · Reviewer_2T1d · 2024-11-04

**Soundness:** 1
**Presentation:** 1
**Contribution:** 1
**Rating:** 3
**Confidence:** 5

**Summary:**

This paper introduces an agent for drug-target interaction task, which uses a basic ReAct framework with access to search, KG and binding affinity prediction models. The tool design and the agent frameworks are rather simple and straightforward. The results evaluated on BindingDB pKd scores show that the proposed agent is better than plain GPT-4, and the effectiveness of each component.

**Strengths:**

* This work could be a starting point for further investigation into drug discovery agent
* The proposed evaluation setup could be used by future works for agent usage evaluation on the drug-target interaction prediction task.

**Weaknesses:**

* The novelty is very limited. The method (tools design and agent design) is straightforward, the proposed agent is more like a small-scale engineering efforts
* Evaluation is not sound. The scale of 10 test cases is too limited, and there is no evaluation with non-agent baselines (such as existing methods for DTI prediction)
* Related works are missing: many works exist about agents for scientific discovery, it would be nice to reference them and discuss the difference
* Some design choices are lack of sufficient justification or empirical evidence (please see the questions section below)

**Questions:**

* Why the KG, Search, and AI agents are designed to be separate agents? Their functions are rather simple, which can be totally achieved as tools bound to the main agent. What is the necessity to use a separate agent?
* In the workflow section, what is the difference between step 4 and 5? Seems both involve getting individual scores of each method and merging the scores.
* What is the system role of the three agents: KG, search and AI? What would be the exact input and output expected for these agents?

---

### Official Review · Reviewer_ChGk · 2024-11-05

**Soundness:** 2
**Presentation:** 4
**Contribution:** 1
**Rating:** 3
**Confidence:** 4

**Summary:**

This paper introduces a multi-agent framework to improve LLM-based reasoning for the task of drug-target interaction and, supposedly, repurposing. The approach encompasses three main "agents" - pieces of code that call a DTI ML model, query a KG and conduct a google search query. These agents are instantiated by am "integrator agent", their results are post-processed given a hard-coded set of rules and combined with a hard-coded formula.

The experimental results seem to indicate that the overall method greatly improves over GPT4 and there are some merits of combining the three "agents", i.e. removing each one of them seems to negatively impact the scores.

**Strengths:**

The (perhaps unsurprising) finding that the proposed framework is better than GPT-4 is a good thing.

The presentation of the paper is clear - it's clear and easy to read and understand.

The commitment to open science is laudable.

**Weaknesses:**

My main concern with this paper is that I fail to see a major contribution. Let me clarify:

While the overall results seem promising, they indeed stem from combining the hard-coded scripts, mainly from the trained ML model and to a lesser extent from mining a curated KG, rather than performing any (autonomous) reasoning per se.
More specifically, it is not clear to me what the contribution of LLMs is specifically - all the interaction with external tools is hard-coded and the combination of the final scores is also pre-defined. The more interesting parts from AI/NLP perspective, the KG agent and the search agent, could have used some level of interaction, in order to find dynamic parameters for scoring the paths and/or search results, respectively, but they do not do so - the weighting of the paths in the KG is hard-coded and the scoring of the google search results is based on keyword matching. Motivation and rationale for choosing these exact weights is missing.

Furthermore, I find it a little odd that the statistical significance results are only reported for comparisons between GPT4 and the full system, comparisons between the full system and the ablated agents are missing, suggesting that they're indeed not significant and performance is likely driven by the ML model predictions (unsurprisingly so).

Minor bits concern exaggerations, for example section 3.2 claims to use ReAct and LEAST-TO-MOST reasoning methods, yet I fail to find their trace in the supplemented code. Furthermore, I am not sure about the use of the term "agent" where all the "agents" do are pre-determined sequences of actions such as calling hard-coded scripts or combining values using a pre-defined formula. Similarly the concluding remark in the introduction is certainly true:  "The framework we developed, although initially designed for biological applications, can be adapted to various other fields requiring multi-perspective." - but only if the person adapting the framework is willing to write and integrate the python tools which do all the work.

Overall, I think this could make a nice demonstration paper, but I doubt the contribution is major enough to warrant acceptance at ICLR.

**Questions:**

Can you please elaborate on the contribution of AI agents here. How would the results differ if I just obtained the scores using the three python scripts and then combine them using the formula described in appendix A.2?

---

### Official Review · Reviewer_27mZ · 2024-11-08

**Soundness:** 2
**Presentation:** 2
**Contribution:** 2
**Rating:** 3
**Confidence:** 3

**Summary:**

This paper presents DrugAgent, a multi-agent LLM system for drug-target interaction prediction that combines three specialized agents: an AI agent for ML predictions, a KG agent for knowledge graph retrieval, and a search agent for web-based information retrieval. The system employs a coordinator agent to integrate results using a linear combination of prediction scores (Smerged = αSAI + βSKG + γSSearch). The authors evaluate their system on the BindingDB dataset and report improved performance compared to GPT-4 baseline.

**Strengths:**

1. Integration of heterogeneous data: The agent framework attempts to combine different information sources and prediction models as different agents, providing analysis from multiple perspectives.
2. Practical application: The work includes case studies showing how the developed system can be applied to analyze real drug target connections.

**Weaknesses:**

1. Limited baseline comparison: The main evaluation only compares with simple prompting on GPT-4 and model ablations (without AI/KG/Search agent). Given the multi-modal nature of the database (KG and textual literature), comparisons with traditional KG and NLP link prediction methods, as well as RAG models, are notably missing.

I suggest the authors improve the soundness of the paper by including comparisons with 1) traditional ML methods 2) traditional KG methods 3) RAG models 4) other multi-agent systems from the general domain.

2. Questionable "multi-agent" framework:
    - The current design appears to be a simple linear combination of different models' results rather than a true multi-agent system.
    - The role and function of the coordinator agent is unclear.
    - The weights of the linear function are determined practically rather than through principled agent collaboration.

I suggest the authors consider more flexible ways of agent collaboration and think about what is the key advantage of a 'multi-agent' system vs a simple combination of different models' result. For example:
- Each agent could output their reasoning process and confidence score
- The coordinator agent could dynamically decide the final score based on these outputs, instead of using a human-defined linear combination
- Implement inter-agent communication instead of having each agent make predictions solely based on its own database and method

Additionally, we can find form the current discussion of the case study that, the current score from each model is like a black box. Adding intermediate reasoning processes as above would also help humans understand the predictions better.

**Questions:**

Could the author elaborate more what is the main different between the proposed 'multi-agent' system to a simple combination of different model's prediction results? What is the role and function of the coordinate agent? What is its system prompt and received message?

---

### Meta-Review · Area_Chair_szb7 · 2024-12-15

**Metareview:**

Summary:

This paper presents a multi-agent LLM system for predicting drug-target interaction, which combines three specialized agents: an AI agent for ML predictions, a KG agent for knowledge graph retrieval, and a search agent for web-based information retrieval, and then employs a coordinator agent to integrate the three agents’ results through a linear combination.  The paper shows that the proposed system achieves better performance than GPT-4 on one dataset (BindingDB) and conducts an ablation study to show the contributions of each agent. All reviewers have pointed out similar major weaknesses, which are summarized as follows.

Strengths:

Overall, reviewers find that the presentation of the paper is clear, and the integration of multiple specialized agents that leverage heterogeneous data is a contribution and could serve as a starting point for investigating drug discovery agents in the future.

Weaknesses:

1. The proposed multi-agent framework does not demonstrate much collaboration and interaction among multiple agents (which are typically key features of a multi-agent system), but simply a linear combination of different models. Hence calling the system “multi-agent” does not seem appropriate, and the technical contribution is very limited.

2. Comparison with baselines is very limited. The current experiments are limited to comparison with GPT-4 and system ablations (without a certain agent). The paper should include comparison with state-of-the-art methods for this task (including those using one specific data source).

**Additional Comments On Reviewer Discussion:**

The authors did not provide a response to the reviews.

---

### Decision · Program_Chairs · 2025-01-22

Reject